# Wildfire Risk Assessment in Liangshan Prefecture, China Based on An Integration Machine Learning Algorithm

**Lingxiao Xie** [1], **Rui Zhang** [1,2,*], **Junyu Zhan** [1], **Song Li** [1], **Age Shama** [1], **Runqing Zhan** [1], **Ting Wang** [1], **Jichao Lv** [1], **Xin Bao** [1] and **Renzhe Wu** [1]

1 Faculty of Geosciences and Environmental Engineering, Southwest Jiaotong University, Chengdu 610031, China
2 State-Province Joint Engineering Laboratory of Spatial Information Technology of High-Speed Rail Safety, Southwest Jiaotong University, Chengdu 610031, China
* Correspondence: zhangrui@swjtu.edu.cn

**Abstract:** Previous wildfire risk assessments have problems such as subjectivity of weight allocation and the linearization of statistical models, resulting in generally low robustness and low generalization ability of fire risk assessment models. Therefore, in this paper, we explored the potential of integration machine learning algorithms to build wildfire risk assessment models. Based on analyzing fire data's spatial and temporal distribution, we selected 10 triggering factors of topography, meteorology, vegetation, and human activities, using frequency ratio (FR) to provide uniform data representation of triggering factors. Next, we used the Bayesian optimization (BO) algorithm to perform hyperparametric optimization solutions for various machine learning models: support vector machine (SVM), random forest (RF), and extreme gradient boosting (XGBoost). Finally, we constructed an integration machine learning algorithm to acquire a fire risk grading map and the importance evaluation corresponding to each triggering factor. For validation purposes, we selected Liangshan Prefecture in Sichuan Province as the specific study area and obtained MCD64A1 burned area product to extract the extent of burned areas in Liangshan Prefecture from 2011 to 2020. The accuracy, kappa coefficient, and area under curve (AUC) were then applied to assess the predictive power and consistency of the fire risk classification maps. The experimental analysis showed that among the three models, FR-BO-XGBoost had the best performance in wildfire risk assessment in the Liangshan region (AUC = 0.887), followed by FR-BO-RF (AUC = 0.876) and FR-BO-SVM (AUC = 0.820). The feature importance result indicated that the study area's most significant effects on wildfires were precipitation, NDVI, land cover, and maximum temperature. The proposed method avoided the subjective weighting and model linearization problems. Compared with the previous methods, it automatically acquired the importance of the triggering factors to the wildfire, which had certain advantages in wildfire risk assessment, and was worthy of further promotion.

**Keywords:** frequency ratio; MCD64A1; Bayesian optimization; support vector machine; random forest; extreme gradient boosting

## 1. Introduction

Forests are vulnerable to and damaged by pests, natural climatic hazards, fires, and other disasters, with fire being the most significant disturbance to forest resources. Forest fires can cause the degradation of vegetation, the death of animals in forest areas, jeopardize people's safety, lead to socio-economic losses, etc. Numerous forest fires occur worldwide every year. Forest fires have become a focus of international attention [1]. China is one of the countries where forest fires are frequent, with an annual average of 12,683 fires and an annual average fire area of 6748 km$^2$ from 1950 to 2010, and the damage rate is higher than the world average [2]. Liangshan Prefecture is a high-risk area for forest fires in Sichuan Province, and in recent years, several forest fires have occurred in the region. In March

2019, 31 people were killed in a forest fire in Muli County [3], Liangshan Prefecture, due to a lightning strike, and in March 2020, 19 people were lost in a forest fire in Xichang City [4]. Frequent forest fires and severe loss of life and property have put Liangshan Prefecture in the spotlight [5]. In general, analyzing the triggering factors of wildfires in this region, constructing a wildfire risk assessment model, and scientifically conducting wildfire risk assessment are essential for fighting and preventing fires and establishing a wildfire prevention system.

Constructing a wildfire risk assessment model requires wildfire occurrence data and wildfire triggering factor data. The satellite remote sensing technology uses multi-spectral, microwave, and multi-source sensor systems to efficiently carry out space-based earth observation and land surface information acquisition. It has the outstanding technical advantages of a comprehensive monitoring range, high spatial and temporal resolution, efficient response, and irreplaceable benefits in obtaining wildfire triggering factors and wildfire occurrence data. Domestic and foreign scholars have conducted relevant research on wildfire risk assessment, such as factor-weighted superposition, hierarchical analysis, and machine learning algorithms. Yang Congrui et al. [6] took Shangri-La, the core area of the three rivers (Jinsha, Lancang and Nujiang rivers), as the study area, and selected vegetation type, topographic data and proximity to residential areas as the primary forest fire triggering factors, and used the factor-weighted overlay method to classify forest fire risk levels. Zhao Pengcheng et al. [7] took the Laoshan National Forest Park in Nanjing as the study area, selected meteorological data, topographic data and proximity to residential areas and roads as triggering factors, and used the analytic hierarchy process (AHP) to classify forest fire occurrence levels. Deng O. et al. [8] constructed a logistic forest fire risk model based on moderate resolution imaging spectroradiometer (MODIS) data and forest fire triggering factors to study the forest fire risk zoning in Heilongjiang Province. Huang Baohua et al. [9] analyzed the causes of forest fires in Shandong using MODIS remote sensing images with topography, vegetation, and weather data, and established a binomial logistic regression model based on 15 explanatory variables affecting the occurrence and non-occurrence of forest fires to estimate the probability of explanatory variables and the event of forest fires. Jaiswal et al. [10] combined India Remote-Sensing Satellite (IRS) 1D LISS III remote sensing data with topographic and other statistical data to assign weights to fire triggering factors and mapped forest fire risk distribution based on a geographic information system (GIS) in Madhya Pradesh, India. Pourtaghi et al. [11] compared the performance of boosted regression tree (BRT), generalized summation model (GAM), and random forest (RF) in forest fire risk assessment in Minudasht County, Golestan Province, Iran, and showed that BRT has better accuracy than GAM and RF. Ljubomir Gigović et al. [12] used SVM, RF and integrated models to assess the forest fire risk in Tara National Forest Park, Serbia, and the results showed that the integrated model has a good prediction accuracy.

However, a simple factor overlay method cannot accurately reflect the influence of fire triggering factors on fire occurrence and fails to calculate the probability of fire risk accurately. A hierarchical analysis is more subjective and suffers from the deficiency of uncertainty in evaluating the situation, and logistic regression only describes the linear relationship between the triggering factors, which suffers from poor robustness and accuracy. On the other hand, machine learning algorithms achieve better performance in wildfire risk assessment. They can reasonably deal with the complex non-linear relationship between fire occurrence and triggering factors. But the lack of proper representation of input data for a single algorithm may lead to models that do not correctly represent the actual spatial distribution of the sample set. Hybrid models are an effective solution to this problem [13]. Their advantage is that the fire location and the weights of different categories of fire triggering factors can be obtained through statistical model analysis. They can also provide proper data representation for machine learning input. On this basis, the complex non-linear relationship between the various triggering factors is revealed through

the machine learning model, and an accurate wildfire risk assessment model is finally obtained which can objectively reflect the influence degree and contribution of each factor.

In order to solve the problems of subjectivity of weight allocation and linearization of statistical models in existing models, this paper proposed an integrated machine learning algorithm and applied it to the Liangshan Prefecture in Sichuan Province. In this case, machine learning algorithms were integrated through frequency ratios to obtain a wildfire risk assessment model. To further reveal the influence of triggering factors on wildfires, the main triggers of wildfires in the region were obtained based on the feature importance of the optimal model. Meanwhile, to study the reliability of the models, the accuracy, kappa coefficient, and AUC were used to evaluate the performance of the three models. The related research results can guide wildfire prevention and control management and wildfire genesis in the Liangshan region.

## 2. Materials and Methods

### 2.1. Study Area

Located in the southwest of Sichuan Province, Liangshan Prefecture is situated between $100°15'{\sim}103°53'$ E and $26°03'{\sim}29°27'$ N, with a total area of 60,400 km$^2$ and a total resident population of 4,874,000. The terrain is high in the northwest and low in the southeast, with large surface undulations and rugged terrain. Liangshan Prefecture has a subtropical monsoon climate with distinct dry and wet conditions. The dry season is from November to April, when the territory has plenty of sunshine, and the rainy season is from May to October, when the territory is cool, wet and rainy. The forest area of Liangshan Prefecture is 31,557 km$^2$, with 52.37% forest coverage and 340 million m$^3$ of forest stock volume in the territory. The vegetation types in the territory are diverse, mainly coniferous and broad-leaved forests, with Yunnan pine widely distributed in the central and southern dry-heat valleys, fir and spruce in the northwestern part at higher altitudes, and dense complex, broad forests in the northeastern region.

Liangshan Prefecture belongs to a forest fire-prone area, which is highly susceptible to forest fires due to high temperatures and low precipitation in winter and spring. Several forest fires have occurred in recent years, including two forest fires in Muli (31 people lost their lives and the burned area was about 20 hectares) and Xichang (19 people lost their lives, 3 people were injured, the burned area was 3047.7805 hectares, 791.6 hectares of forest area was damaged and the direct economic loss equaled 97.312 million yuan) that caused significant casualties and economic losses. Therefore, conducting a wildfire risk prediction and assessment in this area is vital. Figure 1 shows an overview of the study area.

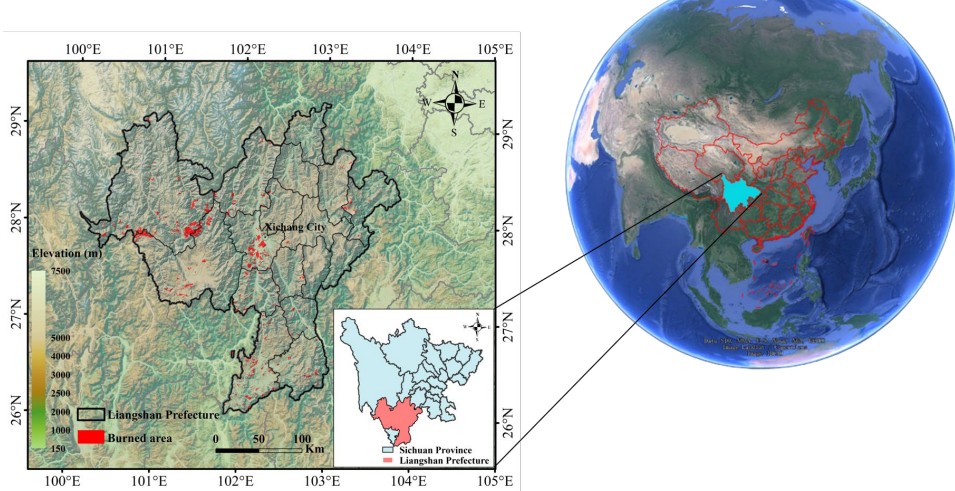

**Figure 1.** Overview of the study area.

*2.2. Data Source*

2.2.1. Fire Database

The MCD64A1 burned area product is a monthly product of burned area mapping with a spatial resolution of 500 m provided by MODIS, which has been proven to be highly accurate [14]. This dataset records the location information, occurrence date, end date, fire area, and confidence level of regional fires. The MCD64A1 burned area data of the Liangshan region from 2011–2020 were obtained from NASA's Terrestrial Data Distribution Center (https://search.earthdata.nasa.gov/search (accessed on 22 April 2022)), and the ten years of burned area data were overlaid. A total of 5951 fire points were extracted.

2.2.2. Selecting Fire Triggering Factors

Selecting fire triggering factors is the first step in fire risk assessment. Many factors, mainly topography, meteorology, vegetation, human activities [5] and accessibility [15], lead to wildfires. Based on the actual situation in Liangshan Prefecture and considering the availability, accuracy, and scientificity of data, ten fire triggering factors were finally selected and classified into four categories (Table A1). Four topographic triggering factors [16] (elevation, slope, aspect and topographic wetness index (*TWI*)) were acquired from the GDEM DEMv3.0 data, respectively. Among them, the formula for calculating *TWI* [17,18] is:

$$TWI = ln(\frac{SCA}{tan\ slope})$$ (1)

*SCA* (specific catchment area) represents the confluence area per unit contour length at any point of the flow across the slope and the slope in the formula is measured in degrees.

Meteorological factors are important factors that trigger wildfires [19]. The meteorological raw raster data were obtained from the National Centre for Atmospheric Sciences [20]. We used interpolation [21] to obtain the average temperature, maximum temperature, and average precipitation data in the study area for the last ten years to get higher resolution meteorological data.

Land cover data were obtained from GlobeLand30 data [22]. It is the first global geographic information product provided by China to the United Nations. Normalized difference vegetation index (*NDVI*) [23–26] data were obtained from the MOD13Q1 data product [27,28] and synthesized using the maximum value composite method [29]. The *NDVI* was calculated as follows:

$$NDVI = \frac{NIR - RED}{NIR + RED}$$ (2)

where *NIR* is the near-infrared band and *RED* is the red band.

Human activities are also a causal factor and a significant evaluation component of wildfires [30]. Population density reflects the level of human activities. The population density data were obtained from the World pop population dataset [31] in Google Earth Engine (GEE) [32].

To meet the modeling needs and ensure the accuracy of the data, we uniformly converted the data to Albers isometric projection. The raster layers of each triggering factor were uniformly resampled to 120 m resolution. They were then were reclassified with a related literature study [33]. (A grading diagram of each fire triggering factor is provided in Figure A1).

*2.3. Technology Route*

Figure 2 shows the workflow of this paper. Following the extraction of fire points for the last decade, a buffer zone with a radius of 1 km was then established on the basis of the fire points, centered on a single fire point. Outside the buffer zone, an equal amount of non-fire point data was selected in combination with land use types. Calculating the frequency ratio values of 10 triggering factors, we extracted the frequency ratio values of different fire points and non-fire points using the extract muti values to point tool [34]. We

analyzed the correlation between each triggering factor using multicollinearity analysis [35]. Subsequently, we used random selection and cross-validation methods to divide the entire samples into a training data set (70%) and validation data set (30%). They were then inputted into the SVM, RF, and XGBoost models, and the Bayesian optimization algorithm was selected to optimize the hyperparameters of the three models. Finally, utilizing accuracy, kappa coefficient, and AUC [36] to evaluate the prediction performance of the model, the trained model was used to predict and risk map the entire Liangshan region, and the study area was classified into five classes using the natural breaks (Jenks) method [37]: very low, low, medium, high, and very high. The importance of the triggering factors was then ranked in conjunction with the optimal model.

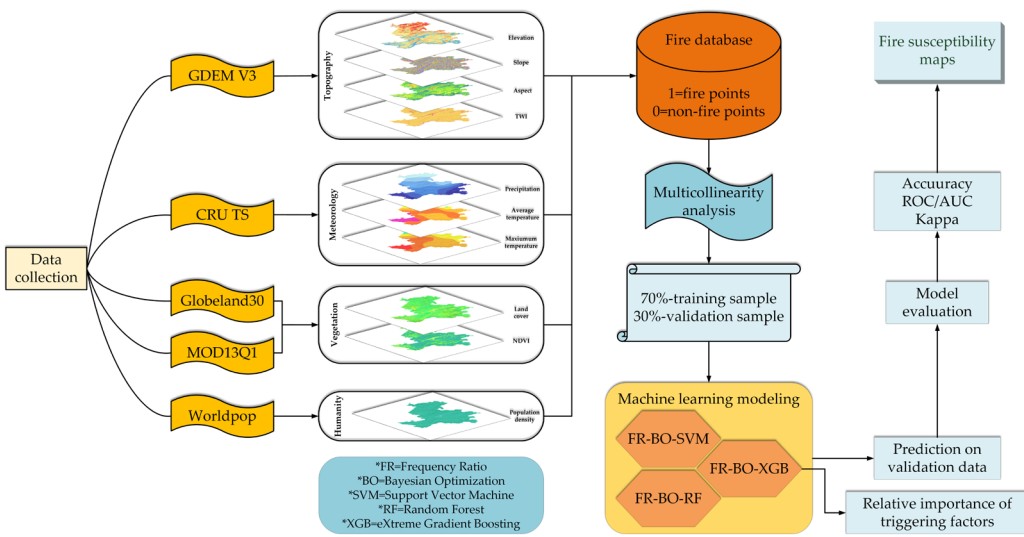

**Figure 2.** Technology route.

## 2.4. Multicollinearity Analysis

Tolerance level (*TOL*) and variance inflation factor (*VIF*) are now commonly used to predetermine the relationship between the triggering factors. It is generally accepted that the presence of multicollinearity is indicated when *TOL* < 0.2 and *VIF* > 4 [13,38]. Among them, *TOL* and *VIF* are calculated as follows:

$$TOL = 1 - R^2 \tag{3}$$

$$VIF = \frac{1}{1 - R^2} = \frac{1}{TOL} \tag{4}$$

where $R^2$ is the coefficient of complex determination.

## 2.5. Fire Risk Assessment Model Construction

### 2.5.1. Frequency Ratio

*FR* is a binary statistical model widely used to study the riskiness of various natural hazards [13,39–42]. The main advantage of this model is its ease of implementation, which allows the calculation of weights for each category based on the spatial relationship between the location of the fire point and the fire triggering factors. The formula for *FR* is:

$$FR = \frac{\frac{Np(LX_i)}{\sum_{i=1}^{m} Np(LX_i)}}{\frac{Np(X_j)}{\sum_{j=1}^{m} Np(X_j)}} \tag{5}$$

where, *FR* denotes the frequency of fires in category *i* of the triggering factor, $Np(LX_i)$ is the number of fire points in classes *i* of the triggering factor *X*, and $Np(X_j)$ is the number of

pixels in classes *j* of the triggering factor *X*. M is the total number of classes in the triggering factor $X_i$.

### 2.5.2. Bayesian Optimization Algorithm

The Bayesian optimization (BO) algorithm was proposed by Pelikan [43] in 2002, which is a very effective global optimization algorithm to find the optimal global solution. This study used the Bayesian optimization algorithm to determine the hyperparameters of different fire risk models (SVM, RF, XGBoost). It uses the Gaussian process theory model [44], which fully considers the previous parameter information and continuously updates the prior knowledge. The algorithm process consists of three main steps. Step 1 is selecting the next most "promising" evaluation point based on the maximized acquisition function [45]. Step 2 is evaluating the objective function [45] value based on the evaluation point chosen. Step 3 is adding the newly obtained input observation pairs to the historical observation set and updating the probabilistic proxy model for the next iteration. The algorithm flow is as follows (Algorithm 1):

---

**Algorithm 1** Bayesian optimization process.

---

1.  Initialize the hyperparameter vector $X_0$
2.  For t = 1, 2, . . . do
3.  Maximize the acquisition function to obtain the next evaluation point:
    $x_t = argmax_{x \in \chi}\alpha(x|D_{1:t-1})$
4.  Evaluate the value of the objective function $y_t = f(x_t) + \varepsilon_t$
5.  Integrating Data: $D_t = D_{t-1} \cup \{x_t, y_t\}$, and update the probabilistic agent model
6.  End for

---

### 2.5.3. Support Vector Machine

Support vector machine (SVM) is a supervised learning method based on statistical learning theory. It solves non-linear and high-dimensional pattern recognition problems relatively well with fewer samples and has been widely used in natural disaster assessment studies [46–49]. The support vector machine aims to find a hyperplane in the n-dimensional data space. It separates these two classes based on the maximum interval. In this case, (0,1] was introduced to consider the issue of misclassification. In addition, Vapnik [50] oriented kernel functions to account for non-linear decision boundaries. Chong Xu et al. [46] found that the kernel function of the support vector machine model performs best in risk assessment when a Gaussian kernel function is selected. Therefore, the kernel function was chosen as the Gaussian function in this study.

### 2.5.4. Random Forest

Random forest (RF) is a nonparametric supervised learning method applied to classification and prediction, first proposed by Breiman [51] in 2001. It is one of the practical algorithms of the Bagging integration strategy [52]. A random forest is a classifier that contains multiple decision trees and trains each decision tree with mutually independent data. The final prediction is then obtained by voting or taking the average.

Random forest uses the bootstrap self-service method for resampling, which generates a new training set by randomly selecting n (generally 2/3 of the whole sample set) samples from the entire collection with put-back. It constructs mutually independent decision trees by training the new sample set and combines the trained n decision trees into a forest. Each of these trees has the same distribution. The classification error depends on the classification ability of each tree and the correlation between them. The remaining unextracted data set called out of bag (OBB) error is an unbiased estimate that can verify the model's performance and prevent overfitting. The random forest generalization error is defined as:

$$P^* \leq \frac{\rho(1-s^2)}{s^2} \tag{6}$$

where $\rho$ is the average correlation of the decision tree and $s$ is the average intensity of the decision tree.

### 2.5.5. eXtreme Gradient Boosting

eXtreme gradient boosting (XGBoost) is a novel gradient boosting decision tree (GBDT) algorithm that was proposed by Chen and Guestrin in 2016 [53]. The XGBoost model uses Taylor's second-order expansion to optimize the loss function. It supports CPU multi-threaded parallel computing and adds a regular term to the loss function, making its computing efficiency and generalization ability significantly better than that of other machine learning algorithms. The XGBoost model is expressed as follows:

$$\hat{y}_i = \sum_{k=1}^{K} f_k(x_i), f_k \in F \tag{7}$$

where $\hat{y}$ denotes the predicted value of the ith sample, $K$ is the number of decision trees, $x_i$ denotes the input data of the ith sample, $f_k(x_i)$ is the $k$th decision tree generated by the $k$th iteration, and $f_k$ is a function in the tree collection space $F$. The objective function is:

$$Obj = \sum_{i=1}^{N} l\left(y_i, \hat{y}_i\right) + \sum_{K=1}^{K} \Omega(f_k) = \sum_{i=1}^{N} l\left[y_i, \hat{y}_i^{t-1} + f_t(x_i)\right] + \sum_{K=1}^{K} \Omega(f_k) \tag{8}$$

In Equation (8), the first part is the loss function, which is used to describe the error between the predicted value and the actual value, and the second part is the standard term, which can effectively control the complexity of the model to build a tree structure model and prevent overfitting [54].

### 2.6. Model Performance Evaluation Methods

Accuracy is a metric used to evaluate classification models and is defined as the percentage of outcomes that are correctly predicted by the model as follows:

$$Accuracy = \frac{TP + TN}{TP + TN + FP + FN} \tag{9}$$

$TP$ is the true positive rate, $TN$ is the true negative rate, $FP$ is the false positive rate, and $FN$ is the false-negative rate.

The kappa coefficient is a metric used for consistency testing. Its expression is as follows:

$$Kappa = \frac{P_0 - P_e}{1 - P_e} \tag{10}$$

where $P_0$ is the consistency of prediction and $P_e$ is the chance consistency. Kappa coefficients can be divided into five groups to indicate different levels of consistency: 0.0 to 0.20 for very low consistency, 0.21 to 0.40 for average consistency, 0.41 to 0.60 for moderate consistency, 0.61 to 0.80 for high consistency, and 0.81 to 1 for almost perfect consistency.

The receiver operating characteristic (ROC) is determined by plotting a set of thresholds or critical values, with the true positive rate (sensitivity) as the vertical coordinate and the false positive rate (1-specificity) as the horizontal coordinate of the curve. The area measures the precision of the results under the curve (ROC). The expressions for the true positive rate (sensitivity) and the false positive rate (1-specificity) are as follows:

$$Sensitivity = \frac{TP}{TP + FN} \tag{11}$$

$$Specificity = \frac{TN}{FP + TN} \tag{12}$$

AUC classifies the performance of prediction models into four categories: 0.5~0.7: low effect, 0.7~0.85: average effect, 0.85~0.95: perfect effect, and 1 indicates an ideal classifier.

*2.7. Feature Importance*

Feature importance [55] is a measure used to assess the usefulness of a feature in the model classification prediction process. The higher the importance of a feature, the more valuable the feature is for the model. The tree model-based machine learning algorithm provides a "feature importance" toolkit, the principle of which is to calculate which feature to select as a splitting point based on the gain of the structure score. The importance of a feature is the sum of its occurrences in all trees, i.e., the more a feature attribute is used to build a decision tree in the model, the higher its importance is. In this paper, the magnitude of the effect of different triggering factors on fire is revealed by ranking the importance of features.

## 3. Results

*3.1. Results of Multicollinearity Analysis*

Table 1 shows the result of multicollinearity analysis; all ten fire triggering factors *TOL* are more significant than 0.2, and *VIF* are less than 4. It indicates no covariance in fire triggering factors, which can be used for subsequent fire risk assessment.

**Table 1.** Results of multicollinearity analysis.

| Num | Factor | TOL | VIF |
|---|---|---|---|
| 1 | Elevation | 0.681 | 1.468 |
| 2 | Aspect | 0.893 | 1.120 |
| 3 | Slope | 0.651 | 1.536 |
| 4 | TWI | 0.664 | 1.506 |
| 5 | Landcover | 0.678 | 1.476 |
| 6 | NDVI | 0.654 | 1.528 |
| 7 | Precipitation | 0.745 | 1.342 |
| 8 | Average temperature | 0.308 | 3.244 |
| 9 | Maximum temperature | 0.389 | 2.572 |
| 10 | Population density | 0.589 | 1.698 |

*3.2. Fire Risk Model Construction*

Since the fire and non-fire samples are in different categories of triggering factors, the corresponding frequency ratios should be assigned according to their classes. The labels of fire points are set as "1", non-fire points are designated as "0", and the final results are used as the model's input data. The input data are divided into a training data set (70%) and a validation data set (30%) using random selection and cross-validation methods. The frequency ratios of the categories of different triggering factors are shown in Table A2.

After assigning the sample points to frequency ratios, the data were trained and tested on the Jupyter Notebook platform using SVM, RF, and XGBoost based on Python. Since the fire risk assessment models' performance depends mainly on the models' hyperparameters, the Bayesian optimization algorithm was used to find the optimal parameters of each model. The optimal parameters of the three models are shown in Table A3.

*3.3. Wildfire Risk Assessment Mapping*

In this section, we trained three machine learning integration algorithms to fit the relationship between each triggering factor and fire to predict the sensitivity index of each image element in the study area. Finally, we acquired the fire risk raster map for the whole study area and reclassified the raster maps by the natural breaks (Jenks) method. The natural breaks method [56] is based on the univariate classification method in cluster analysis. It minimizes the differences in classes while maximizing the differences between classes by calculating the data breakpoints between classes under a certain hierarchy. Its advantage is to make the most effective distinction between similar values in the data, so the natural breaks method was chosen for classification in this study. The maps were reclassified into five classes, namely very low, low, medium, high, and very high (Figure 3). The high

incidence areas of wildfires identified by these three models are mostly concentrated in the south, central and northwest of the region.

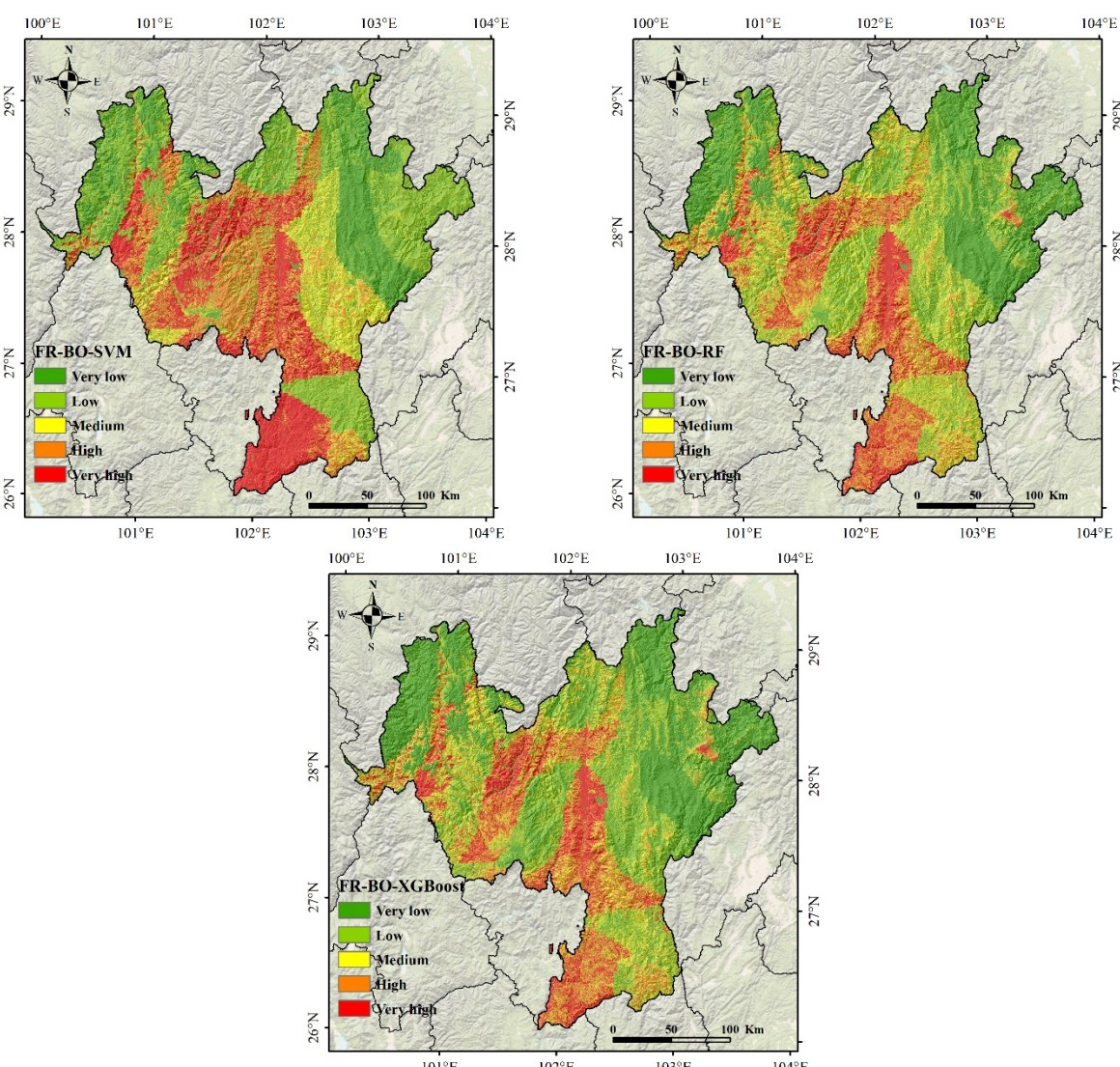

**Figure 3.** Risk assessment mapping results of three machine learning models.

In addition, the percentages of the different risk levels of the three machine learning algorithms were calculated separately. Figure 4 shows that the five risk levels of SVM account for 20.86% (very low), 29.62% (low), 12.82% (medium), 14.15% (high) and 22.54% (very high) respectively, the five risk levels of RF account for 28.69% (very low), 23.80% (low), 18.69% (medium), 16.23% (high), and 12.59% (very high), and the five risk classes of XGBoost account for 34.52% (very low), 23.05% (low), 15.49% (medium), 14.55% (high), and 12.40% (very high).

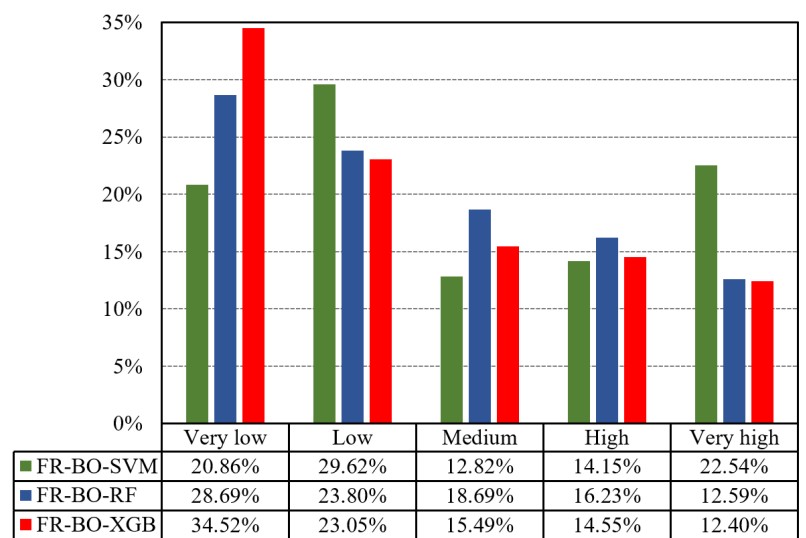

**Figure 4.** Percentage of the risk level of three machine learning algorithms.

*3.4. Model Evaluation*

After successfully constructing the three risk assessment models, the performance of the three models was evaluated using accuracy, kappa coefficient, and area under the ROC curve AUC. The evaluation results of each model are shown in Table 2 and Figure 5.

**Table 2.** Parameters for evaluating the predictive performance of the three models.

|  | Training Data Set | | | Test Data Set | | |
|---|---|---|---|---|---|---|
| **Parameters** | **FR-BO-SVM** | **FR-BO-RF** | **FR-BO-XGB** | **FR-BO-SVM** | **FR-BO-RF** | **FR-BO-XGB** |
| TP | 2701 | 3368 | 3383 | 1210 | 1402 | 1368 |
| TN | 3479 | 3708 | 3687 | 1415 | 1435 | 1520 |
| FP | 713 | 484 | 479 | 344 | 324 | 265 |
| FN | 1438 | 771 | 782 | 602 | 410 | 418 |
| Sensitivity | 0.653 | 0.814 | 0.812 | 0.668 | 0.774 | 0.766 |
| Specificity | 0.830 | 0.885 | 0.885 | 0.804 | 0.816 | 0.852 |
| Accuracy | 0.742 | 0.849 | 0.847 | 0.735 | 0.794 | 0.809 |
| Kappa | 0.483 | 0.699 | 0.697 | 0.471 | 0.589 | 0.617 |

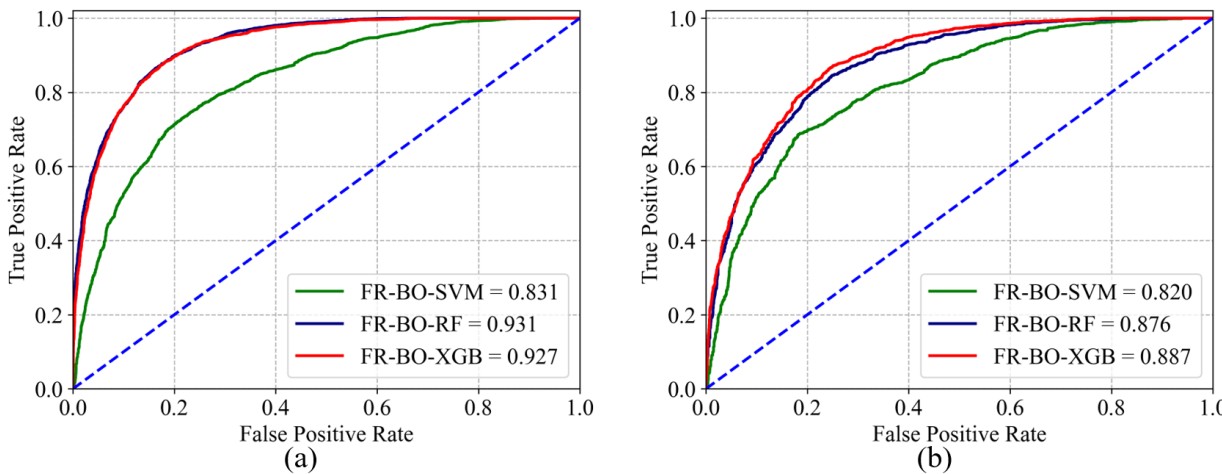

**Figure 5.** (**a**) ROC curves of the success rate of the three models, (**b**) ROC curves of prediction rates of three models.

The training and testing results of wildfire occurrence probability are shown in Table 2 and Figure 5. From the training set, FR-BO-SVM has an average classification effect (AUC = 0.831) and a kappa coefficient of 0.483, which has only moderate consistency, while the other two models, FR-BO-RF and FR-BO-XGB, have perfect classification effects (AUC values of 0.931 and 0.927, respectively) and they both achieve a high degree of agreement with kappa coefficients of 0.699 and 0.697. In addition, the accuracy of FR-BO-SVM (Accuracy = 0.742) is lower than that of the other two models (accuracy = 0.849 and 0.847, respectively). From the test set, the AUC value and kappa coefficient of FR-BO-SVM are 0.820 and 0.471, respectively. It still has an average prediction effect and moderate consistency. FR-BO-RF still has a good prediction effect (AUC = 0.876), but the kappa coefficient is 0.589, which shows medium consistency and accuracy. FR-BO-XGB has demonstrated excellent prediction effectiveness and consistency with AUC and kappa coefficients of 0.887 and 0.617, respectively, and the accuracy remains above 0.8 at 0.809. Combining the above analysis, we can conclude that FR-BO-XGB is the best model among these three models.

### 3.5. Importance of Triggering Factor Features

The model evaluation shows that FR-BO-XGBoost is the best model among the three models, so the model can explain the relationship between fire points and each triggering factor well. However, different triggering factors do not have the same degree of influence on fire, and it is necessary to understand the importance of various triggering factors in starting a fire. The vertical coordinates in Figure 6 are each triggering factors for fire risk assessment. The horizontal coordinates are the ratio of times each feature attribute is used for decision tree node segmentation to the number of times all feature attributes are segmented. The feature importance analysis shows that precipitation is the most important triggering factor for wildfires in Liangshan, followed by NDVI, land cover, and maximum temperature, respectively, while other triggering factors such as elevation, average temperature, and population density have some influence on fire initiation.

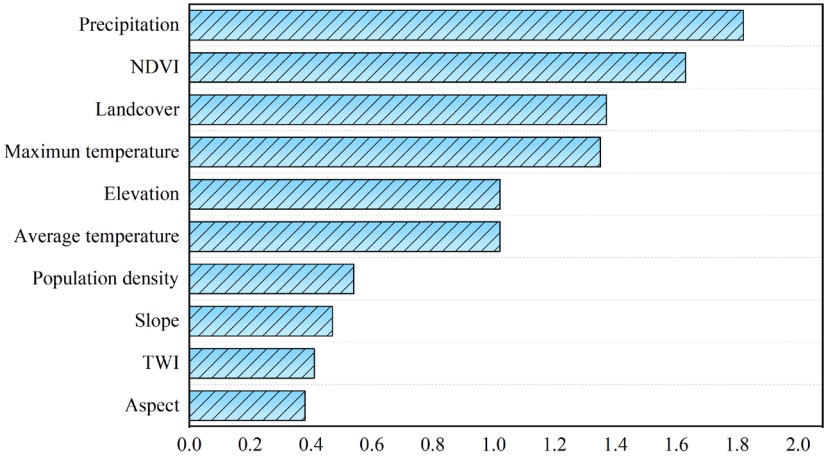

**Figure 6.** Ranking the importance of triggering factor features.

Figure A2 reveals the percentage of the different risk levels of the four main fire initiating factors. We can summarize that the annual precipitation reached more than 800 mm, which was a high incidence area of wildfire (Figure A2a). The role of precipitation on fire is reflected in the fact that higher precipitation means higher fuel loading for brush and grass fires, resulting in higher fire behavior. The influence of NDVI on fire (Figure A2b) was manifested primarily in the fact that the larger the value, the more the high incidence areas of fire, which was mainly related to the rich vegetation types in the Liangshan region. Liangshan is a national critical fly-sown forest area (Figure A2c). The territory is planted with the Yunnan pine as the representative of the flammable tree species. It had been reported that lightning strikes on Yunnan pine caused forest fires in Muli, and land cover

was concentrated on cropland, woodland, grassland, and shrubland, where a large amount of combustible material had accumulated, thus creating the potential for wildfires. The effect of maximum temperature on fires is specifically demonstrated by the fact that higher temperatures lead to lower humidity, which leads to lower fuel moisture, which leads to higher combustibility. Figure A2d demonstrates that above 18 °C is the high incidence of fire in the region.

## 4. Discussion

This paper proposed an integrated machine learning approach to wildfire risk assessment by introducing frequency ratios and machine learning models. The method used frequency ratios to achieve objective weighting of the triggering factors. On this basis we focused on optimizing the machine learning model to reveal the complex nonlinear relationships among the triggering factors, which more objectively reflects the degree of influence and the contribution of each factor. In the final, we achieved a quantitative regional assessment of wildfire risk. For validation purpose, based on the occurrence mechanism of wildfires in the Liangshan region, we selected 10 years of the burned area data (MCD64A1 burned area product), 10 years of meteorological data (precipitation, maximum temperature, average temperature), topographic data (elevation, slope, aspect, TWI), vegetation data (NDVI, landcover) and human activities factor data (population density) to construct a wildfire risk assessment model for the Liangshan region. In the end, we acquired accurate and reliable results.

It was worth noting that topographic factors (elevation, slope, aspect and TWI), meteorological factors (precipitation, average temperature, maximum temperature), vegetation factors (NDVI and landcover) and human activities factors (population density) were all mountain wildfire main triggering factors. However, in different regions and research scenarios, the impact of various factors on fires was quite different. The existing models and methods (such as the factor superposition method, analytic hierarchy process, logistic regression method, etc.) had problems such as subjectivity of weight allocation and linearization of statistical models. As a result, the robustness of the fire risk assessment model was generally not high, and the generalization ability was poor. Aiming at the problem of accurate quantification of the weights of disaster triggering factors, based on the historical fire data in the Liangshan area in the past 10 years, this paper obtained the frequency ratio of each triggering factor through statistical analysis and acquired the weight of the fire location and different categories of each fire triggering factor. On this basis, the complex non-linear relationship between the triggering factors was revealed through the machine learning model, and a wildfire risk assessment model suitable for this research scenario was finally obtained which objectively reflected the influence degree and contribution of each factor.

In previous studies, a risk assessment of forest fires in the Bizerte region of Tunisia in terms of burn index, geographic terrain index, human index, climate index and other indicators using GIS and remote sensing (RS) techniques, Saidi et al. [57]. Zhao, Pengcheng et al. [7] selected topographic, meteorological, vegetation and human factors for forest fire assessment in Laoshan National Forest Park, Nanjing using AHP. Saeedeh et al. [58] utilized a fuzzy analytic hierarchy process (FAHP), the spatial correlation method and the Dong model to predict forest fires in Iran from four aspects: geographic terrain factor, ecological factor, environmental factor and human factor. Elham et al. [59] considered factors such as slope, elevation, and road distance, and applied the analytic network process (ANP) to assess the risk of forest fires in a city. These methods had subjective factor assignments and lack objective consideration of factor weights. The difference in this study was the introduction of frequency ratios to achieve objective weighting of the triggering factors, which avoided the problem of inaccurate quantification of factors. Pan et al. [60] combined a logistic regression model for forest fire assessment in Shanxi Province by selecting elevation, slope, distance from the nearest road, fuel moisture content (FMC), land surface temperature, NDVI and global vegetation moisture index (GVMI), and verified that the AUC of

the model was 0.757. Li Haiping et al. [33] established a logistic risk assessment model for forest fires in Liangshan Prefecture by adopting the main risk factors such as topography, vegetation, meteorology and population density, and finally acquired an AUC of 0.798. The above model only described a simple linear relationship among the factors, causing the model to be poorly generalized and less accurate. The distinction of the machine learning algorithm proposed in this study was that it can solve the complex nonlinear relationship between factors quite well. On this basis, the degree of influence of each triggering factor on fires can be further acquired automatically. It is worth mentioning that, in the identical study scenario, the model performance of the integrated machine learning method for fire risk assessment proposed in this study improved by 0.089 over the former method [33]. This indicates that the method has a promising application in wildfire risk assessment.

## 5. Conclusions

To address the problems of subjective weighting of triggering factors and linearization of models in the current wildfire risk assessment studies, we proposed an integrated machine learning approach to wildfire risk assessment. Frequency ratios were introduced to achieve objective weighting of the triggering factors, and on this basis, the machine learning model was focused on optimizing the model to reveal the complex nonlinear relationships among the triggering factors, which reflected the influence degree and contribution of each factor more objectively. Finally, we achieved the regional quantitative assessment of wildfire risk. For validation purposes, we filtered out 10 wildfire triggers based on the mechanism of wildfires in the Liangshan region. Combined with historical remote sensing fire data, a fire database was constructed and a wildfire risk level map was obtained for the Liangshan region. In the final, the degree of influence of fire triggering factors on wildfires in the region was further revealed in conjunction with the feature importance analysis. The greatest contribution of this study is to overcome the difficulties of subjective selection of factor weights and the linearization of the model in the research process. The findings of the study can provide useful information for the relevant departments to manage and make decisions on wildfire prevention and control in advance. In addition, the method can also be applied to wildfire risk assessment in other regions. It is essential to note that the selection of triggering factors using this method needs to take into account the actual situation of wildfire occurrence in the corresponding area.

However, in this study, predictions were based on a single pixel for each factor layer of the fire occurrence site, ignoring the fact that fire occurrence is also related to the surrounding environment. In future research, deep learning algorithms can be considered to accurately extract the deep features of different factor layers.

**Author Contributions:** Conceptualization, L.X.; Methodology, R.Z. (Rui Zhang); Software, J.Z. and S.L; Validation, A.S. and T.W.; Resources, L.X.; Writing-Original Draft Preparation, R.Z. (Runqing Zhan); Writing-Review & Editing, L.X., J.Z. and S.L.; Visualization, J.L. and R.W.; Supervision, X.B.; Funding Acquisition, R.Z. (Rui Zhang) All authors have read and agreed to the published version of the manuscript.

**Funding:** This research was jointly funded by the National Natural Science Foundation of China (Grant No. 42171355 and 42071410); the Sichuan Science and Technology Program (Grant No. 2019ZDZX0042, 2020JDTD0003, 2020YJ0322, and 2021YFH0038); and the Major Projects of High-Resolution Earth Observation (30-H30C01-9004-19/21).

**Data Availability Statement:** Data is openly available in a public repository.

**Acknowledgments:** The authors would like to acknowledge MCD64A1 burned area data from NASA's Land Data Distribution Center, topographic data from the Geospatial Data Cloud, meteorological data from the National Centre for Atmospheric Sciences, land cover data from the GlobeLand30 data platform, and NDVI and population density data from the GEE platform.

**Conflicts of Interest:** The authors declare that they have no conflict of interest.

## Appendix A

**Table A1.** Data of each triggering factor.

| Category | Triggering Factors | Resolution | Data Source |
|---|---|---|---|
| Topography | Elevation<br>Slope<br>Aspect<br>TWI | 30 m | https://www.gscloud.cn/ (accessed on 23 April 2022 ) |
| Meteorology | Average temperature<br>Maximum temperature<br>Precipitation | 0.5° | https://crudata.uea.ac.uk/cru/data/hrg/ (accessed on 23 April 2022) |
| Vegetation | Land cover<br>NDVI | 30 m<br>250 m | http://www.globallandcover.com/ (accessed on 24 April 2022)<br>https://explorer.earthengine.google.com/ (accessed on 24 April 2022) |
| Humanity | Population density | 100 m | https://explorer.earthengine.google.com/ (accessed on 24 April 2022) |

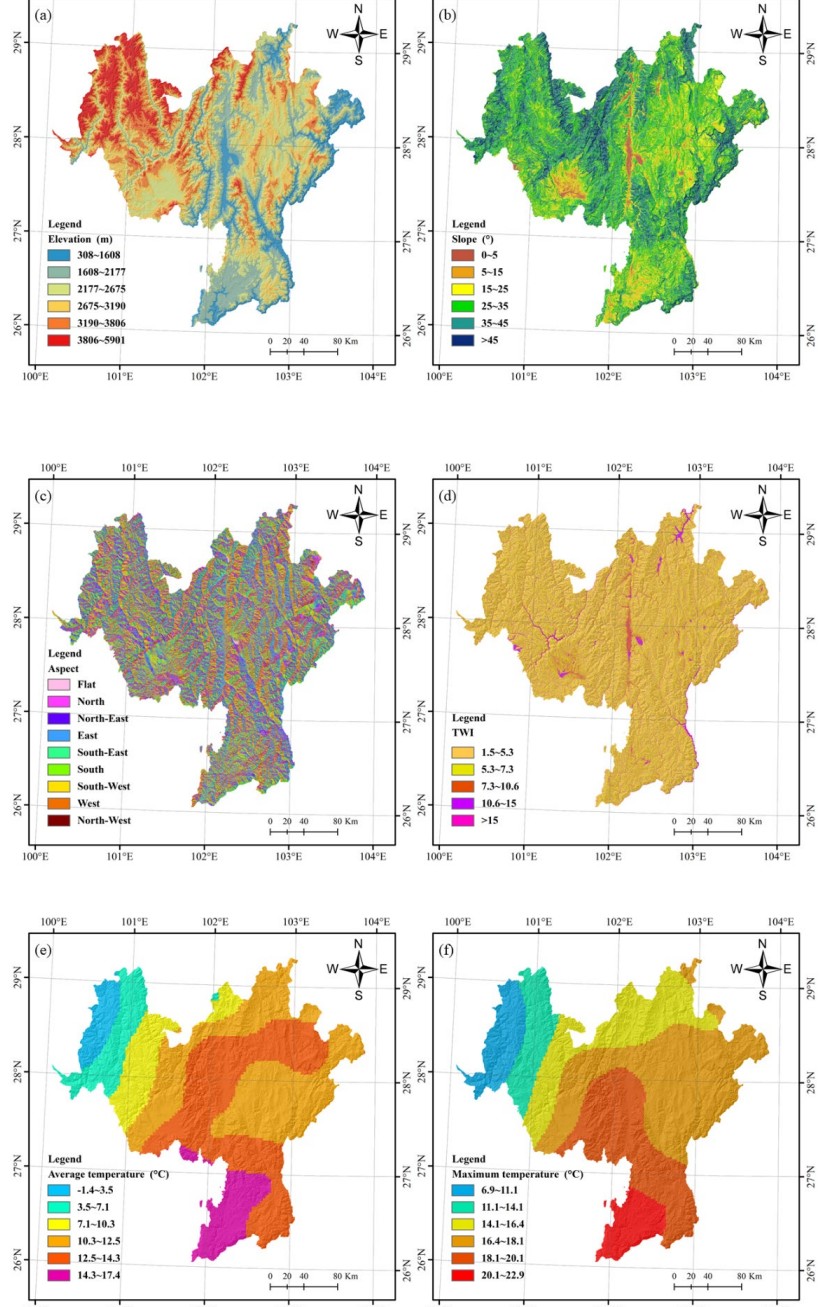

**Figure A1.** *Cont.*

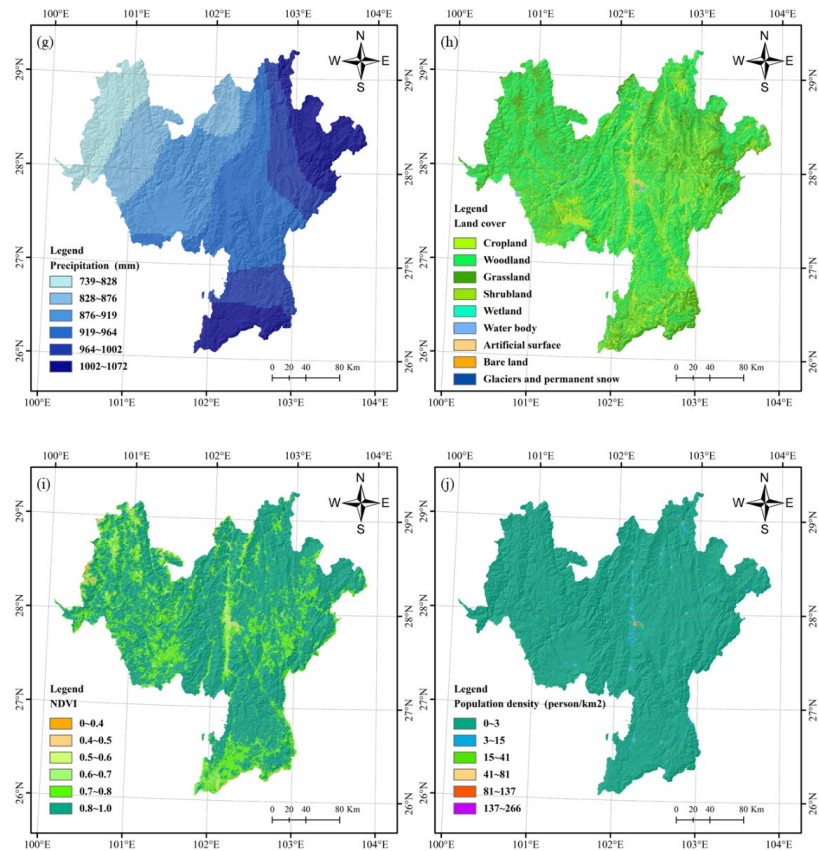

**Figure A1.** Grading diagram of each fire triggering factor; (**a**) Elevation, (**b**) Slope, (**c**) Aspect, (**d**) TWI, (**e**) Average temperature, (**f**) Maximum temperature, (**g**) Precipitation, (**h**) Land cover, (**i**) NDVI, (**j**) Population density.

**Table A2.** Frequency ratios of different triggering factor categories.

| Triggering Factors | Classes | Classes Pixels | Fire Pixels | FR |
|---|---|---|---|---|
| Elevation (m) | 308~1608 | 378,249 | 639 | 1.19 |
| | 1608~2177 | 839,922 | 1422 | 1.19 |
| | 2177~2675 | 1,038,686 | 1237 | 0.84 |
| | 2675~3190 | 926,171 | 1637 | 1.24 |
| | 3190~3806 | 626,194 | 962 | 1.08 |
| | 3806~5901 | 373,870 | 54 | 0.10 |
| Slope (°) | 0~5 | 74,637 | 244 | 2.30 |
| | 5~15 | 155,138 | 197 | 0.89 |
| | 15~25 | 983,516 | 1220 | 0.87 |
| | 25~35 | 1,413,834 | 1952 | 0.97 |
| | 35~45 | 1,125,384 | 1795 | 1.12 |
| | >45 | 430,583 | 543 | 0.89 |
| Slope (°) | 0~5 | 74,637 | 244 | 2.30 |
| | 5~15 | 155,138 | 197 | 0.89 |
| | 15~25 | 983,516 | 1220 | 0.87 |
| | 25~35 | 1,413,834 | 1952 | 0.97 |
| | 35~45 | 1,125,384 | 1795 | 1.12 |
| | >45 | 430,583 | 543 | 0.89 |

**Table A2.** *Cont.*

| Triggering Factors | Classes | Classes Pixels | Fire Pixels | FR |
|---|---|---|---|---|
| Aspect | Flat | 4781 | 1 | 0.15 |
| | North | 447,717 | 477 | 0.75 |
| | Northeast | 505,581 | 506 | 0.70 |
| | East | 603,767 | 722 | 0.84 |
| | Southeast | 572,515 | 884 | 1.09 |
| | South | 480,907 | 885 | 1.29 |
| | Southwest | 509,576 | 945 | 1.30 |
| | West | 555,006 | 880 | 1.11 |
| | Northwest | 503,242 | 651 | 0.91 |
| TWI | 1.5~5.3 | 2,050,828 | 3106 | 1.06 |
| | 5.3~7.3 | 1,406,009 | 1904 | 0.95 |
| | 7.3~10.6 | 482,052 | 647 | 0.94 |
| | 10.6~15 | 203,487 | 246 | 0.85 |
| | >15 | 40,716 | 48 | 0.83 |
| Average temperature (°C) | −1.4~3.5 | 195,154 | 16 | 0.06 |
| | 3.5~7.1 | 373,851 | 705 | 1.33 |
| | 7.1~10.3 | 423,180 | 395 | 0.66 |
| | 10.3~12.5 | 1,466,289 | 2030 | 0.97 |
| | 12.5~14.3 | 1,353,512 | 1930 | 1.00 |
| | 14.3~17.4 | 371,106 | 875 | 1.66 |
| Maximum temperature (°C) | 6.9~11.1 | 333,928 | 207 | 0.44 |
| | 11.1~14.1 | 367,464 | 714 | 1.37 |
| | 14.1~16.4 | 780,108 | 581 | 0.52 |
| | 16.4~18.1 | 1,507,804 | 1993 | 0.93 |
| | 18.1~20.1 | 910,210 | 1697 | 1.31 |
| | 20.1~22.9 | 283,578 | 759 | 1.88 |
| Precipitation (mm) | 739~828 | 447,911 | 246 | 0.39 |
| | 828~876 | 575,927 | 875 | 1.07 |
| | 876~919 | 848,386 | 1995 | 1.65 |
| | 919~964 | 961,536 | 1607 | 1.17 |
| | 964~1002 | 644,965 | 354 | 0.39 |
| | 1002~1072 | 704,367 | 874 | 0.87 |
| NDVI | 0.1~0.4 | 5716 | 6 | 0.74 |
| | 0.4~0.5 | 10,993 | 19 | 1.21 |
| | 0.5~0.6 | 36,355 | 91 | 1.76 |
| | 0.6~0.7 | 142,139 | 391 | 1.93 |
| | 0.7~0.8 | 955,839 | 2243 | 1.65 |
| | 0.8~1 | 3,032,050 | 3201 | 0.74 |
| Landcover | Cropland | 810,215 | 752 | 0.65 |
| | Woodland | 2,418,827 | 3971 | 1.15 |
| | Grassland | 679,862 | 747 | 0.77 |
| | Shrubland | 205,254 | 402 | 1.38 |
| | Wetland | 1638 | 0 | 0.00 |
| | Water body | 35,807 | 0 | 0.00 |
| | Artificial surface | 29,183 | 78 | 1.88 |
| | Bare land | 1654 | 1 | 0.42 |
| | Glaciers and permanent snow | 652 | 0 | 0.00 |
| Population (person/km$^2$) | 0~3 | 4,088,035 | 5653 | 0.97 |
| | 3~15 | 78,498 | 244 | 2.18 |
| | 15~41 | 11,862 | 46 | 2.73 |
| | 41~81 | 2786 | 4 | 1.01 |
| | 81~137 | 1481 | 4 | 1.90 |
| | 137~266 | 430 | 0 | 0.00 |

**Table A3.** Optimal parameters of the three models.

| Algorithm | Parameters |
| --- | --- |
| SVM | Kernel:rbf<br>C:100<br>Gamma:0.1 |
| RF | N_estimators:248<br>Criterion: Gini<br>Max_depth:14<br>Max_features:0.432<br>Min_samples_spilt:14 |
| XGBoost | Max_depth:9<br>Learning_rate:0.051<br>N_estimators:119<br>Min_child_weight:1<br>Subsample:0.827<br>Colsample_bytree = 1<br>Booster:gbtree |

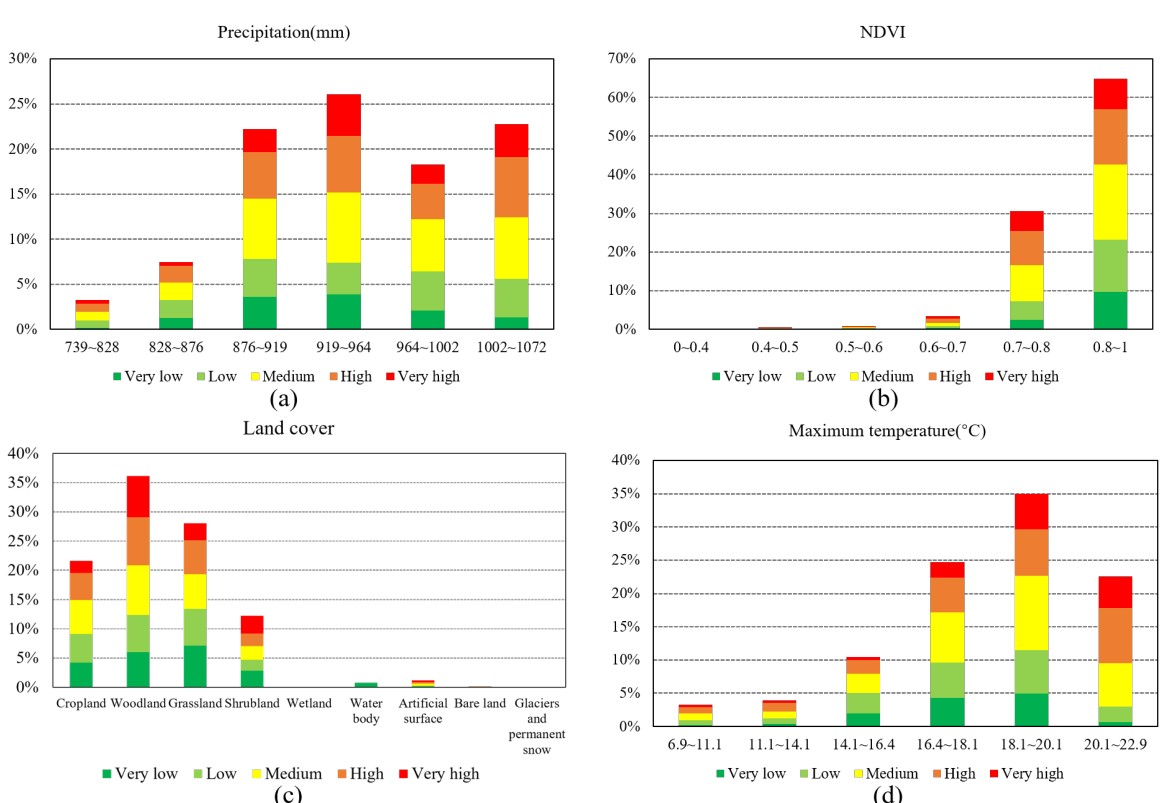

**Figure A2.** Main triggers of wildfires (**a**) Precipitation, (**b**) NDVI, (**c**) Land cover, (**d**) Maximum temperature.

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
