# Peer review of "Wildfire Risk Assessment in Liangshan Prefecture, China Based on An Integration Machine Learning Algorithm"

_remotesensing, doi:10.3390/rs14184592_

Round 1

Reviewer 1 Report (Previous Reviewer 2)

Figure 1 legend must be improved. The elevation must be cited in the text, remove DEM, add m.a.s.l. Red in fires must be more intense. Indicate in the text that this is refered to Liangsham prefecture in Sichuan Province. The other Figure is irrelevant, made smaller (put in the right), and instead must showed all the country (China), and just indicate the study area. Reduce the text in the figures (e.g. not repeat the lat/long).

Table 1 can be in the text or move to one Annex.

Figure 2 must be improved in quality. There must be converted more graphical less text. Besides, in this figure you put your results... I am not sure if this is OK... Maybe it is a good graphical abstract... This is not good for “methods”. Revise and re-think.

Discussion: The discussion included summary of your outputs (lines 372-427). This is not a discussion. Remove of reduce as one introductory sentence. Or move to RESULTS section.

Lines 428-455 are OK as a brief discussion.

Conslusions are not conclusions. Rename this section as “Final Remarks”. Otherwise, delete these sentences and write some conclusions (relevant new knowledge derived from your results).

Author Response

Reviewer 2 Report (New Reviewer)

Author Response

Reviewer 3 Report (Previous Reviewer 1)

The authors present an exciting paper that explores the potential of integration of machine learning algorithms to build forest fire risk, assessment models. The manuscript is clear, relevant for the field and presented in a well-structured manner, and scientifically sound. The manuscript’s results are reproducible based on the details given in the methods section. The manuscript is well written and should be of great interest to the readers. However, all figures with charts could be more significant. 

Author Response

This manuscript is a resubmission of an earlier submission. The following is a list of the peer review reports and author responses from that submission.

Round 1

Reviewer 1 Report

The authors present an exciting paper introducing an integration machine learning algorithm to forest fire risk assessment. Based on analyzing fire data's spatial and temporal distribution, they selected 10 triggering factors of topography, meteorology, vegetation, and human activities, using frequency ratio to provide uniform data representation of triggering factors. Next, they used the Bayesian optimization algorithm to perform hyperparametric optimization solutions for various machine learning models (support vector machine, random forest, and XGBoost). Ultimately, they constructed an integration machine learning algorithm to acquire a fire risk grading map and the importance evaluation corresponding to each triggering factor. The manuscript is clear, relevant for the field, well-structured, and scientifically sound. The manuscript’s results are reproducible based on the details given in the methods section. The manuscript is well written and should greatly interest the readers. The only remark is that the conclusion should mention more about their future work. 

Reviewer 2 Report

Title: An Integration Machine Learning Algorithm for Forest Fire Risk Assessment in Liangshan Prefecture, China

Change for: The Challenge to Improve Forest Fire Risk Assessment:  Integration Machine Learning Algorithm for Liangshan Prefecture in China

Revise English, as well as the Style (check authors instructions carefully).

Abstract: Improve the objective presentation, and better define what is new for Science. The abstract is quite technical, and we need to improve the scientific background. At the end, please specify what is new with this method, why is better, and a sentence with recommendations. We need to highlight the advance in the science framework. Those acronyms listed (lines 30 to 36) must be explained in the text. Abstract must be self-explained.

Keywords: Remove those keywords that already appear on title.

Introduction: Line 49, check square numbers.

The acronyms must be self-explained here too.

The introduction listed some weakness of the common risk valuation methods, and introduce the MLIA as the best solution, however, the MLIA are not enough explained, and justify to be the solution. Please, improve these arguments.

Improve the objective, and add some questions to be solved during this research.

Remove the “methods” after objective. Please, try to be more scientific paper and less technical report during the writing.

Methods:

Title 2.2.1. Change for Fire database.

Title 2.2.2. Change for Selecting fire...

Methods presented a lot of “discussion”, remove those sentences, and move to Discussion section. Remember, methods are quite descriptive, just describe what you do. Nothing else. You must reduce all the methods to the half.

Figure 2 move to Appendix.

Subtitles 2 and 3: Mix these subtitles under “Methods”.

Results: Change the title just for “Results”. You must describe your outputs, not use sentences as “as can be seen”. This is a scientific paper.

Table 4 and 5 move to Appendix. They are not useful to understand the outputs, or to justify why this methodology is better than others.

Fig 7 and 9 must be moved to Appendix.

Discussion is completely lack. You must add a Discussion section!!!!! Please, there is NO one reference here! This is just a brief description of the results! Unacceptable!

Conclusions must describe the NEW knowledge generate to the Science. To date, your conclusions is a summary of your results, just like a technical report. Re-write.

Round 2

Reviewer 2 Report

Thanks to improving the draft. Some sections are much better. However, we need to improve some sections before approve the draft:

(a) Please check the style, the draft need to improve with some spaces, titles, etc. In example: (i) Fig 2 and 7 are lack (please, rename the exinting Figures), as well as Tanle 4 is lack (rename the Figures in order). (ii) Name the appendixes as Appendix 1, 2, 3. Remove the names "Figures" or "Tables" in the Appendix.

(b) Discussion is great considering YOUR point of view. You made a good discussion by your side. Now... what happens with the EXISTING knowledge that were already published? In the discussion section you must compare your outputs with the rest of the World. You need to add references... and highlight here the differences of your proposal and the previous ones. This is not a technical report. 

https://www.sjsu.edu/writingcenter/docs/handouts/Discussion%20Section%20for%20Research%20Papers.pdf

(c) The same for Conclusion. Actually still is a "summary" for your outputs. The conclusion must be short, and just describe the NEW KNOWLEDGE generated by the paper... not the results... the NEW KNOWLEDGE. 

Please, read some papers of the Journal, and try to made something similar. If you not attend these details, it is impossible to approve this draft.
